# Concentration Prediction of Polymer Insulation Aging Indicator-Alcohols in Oil Based on Genetic Algorithm-Optimized Support Vector Machines

**DOI:** 10.3390/polym14071449

**Published:** 2022-04-02

**Authors:** Shuyue Wu, Heng Zhang, Yuxuan Wang, Yiwen Luo, Jiaxuan He, Xiaotang Yu, Yiyi Zhang, Jiefeng Liu, Feng Shuang

**Affiliations:** School of Electrical Engineering, Guangxi University, Nanning 530004, China; wsy18996092096@163.com (S.W.); hengzhang_gxu@163.com (H.Z.); wangyuxuan@st.gxu.edu.cn (Y.W.); gxulyw@163.com (Y.L.); 15777120007@163.com (J.H.); yuxiaotang0703@163.com (X.Y.); yiyizhang@gxu.edu.cn (Y.Z.)

**Keywords:** transformer polymer insulation, methanol, ethanol, genetic-algorithm-optimized support vector machine (GA-SVM), concentration prediction

## Abstract

The predictive model of aging indicator based on intelligent algorithms has become an auxiliary method for the aging condition of transformer polymer insulation. However, most of the current research on the concentration prediction of aging products focuses on dissolved gases in oil, and the concentration prediction of alcohols in oil is ignored. As new types of aging indicators, alcohols (methanol, ethanol) are becoming prevalent in the aging evaluation of transformer polymer insulation. To address this, this study proposes a prediction model for the concentration of alcohols based on a genetic-algorithm-optimized support vector machine (GA-SVM). Firstly, accelerated thermal aging experiments on oil-paper insulation are conducted, and the concentration of alcohols is measured. Then, the data of the past 4 days of aging are used as the input feature of SVM, and the GA algorithm is utilized to optimize the kernel function parameter and penalty factor of SVM. Moreover, the concentrations of methanol and ethanol are predicted, after which the prediction accuracy of other algorithms and GA-SVM are compared. Finally, an industrial software program for predicting the concentration of methanol and ethanol is established. The results show that the mean square errors (*MSE*) of methanol and ethanol concentration predictions of the model proposed in this paper are 0.008 and 0.003, respectively. The prediction model proposed in this paper can track changes in methanol and ethanol concentrations well, providing a theoretical basis for the field of alcohol concentration prediction in transformer oil.

## 1. Introduction

The power transformer is an indispensable piece of equipment in power grids, as it is responsible for voltage-level transformation, and its operational status is related to the safe and stable operation of the power grid [1,2,3,4]. The insulating materials inside the transformer are composed of oil and paper/pressboard. Under the action of electrical stress, thermal stress, and mechanical stress, the insulating materials gradually degrade. The aging of insulating oil can be slowed down by changing/filtering oil, while the aging of the paper insulation is irreversible. Therefore, the operating period of the transformer depends on the lifespan of its paper insulation [5,6,7,8,9].

It is universally acknowledged that the degradation of oil-impregnated paper is divided into pyrolysis, hydrolysis, and oxidative degradation [10,11]. During the degradation process of paper insulation, various aging indicators (furfural, methanol, ethanol, carbon oxides, etc.) are generated and dissolved in transformer oil. The concentration of the above indicators can be measured, and its correlation with the aging state of the paper insulation is subsequently established. Carbon oxides are derived from both the degradation of paper insulation and are also produced during long-term oxidation of insulating oil [12], which may interfere with the aging assessment results of insulation. In comparison, aging indicators (furfural, methanol, ethanol) are only generated from the degradation of paper insulation and can be used to characterize the aging state of paper insulation. However, with the widespread use of thermally upgraded Kraft (TUK) paper in transformers, furfural is no longer effective in the aging assessment of the TUK paper. Alcohols (methanol and ethanol), as new types of indicators, will not be limited by the type of insulating paper when utilized for aging assessment. Methanol is regarded as an indicator closely related to the rupture of 1,4-b-glycosidic bonds in cellulose. Additionally, ethanol is a by-product of the degradation of cellulose insulation through levoglucosan as an intermediate [13]. Since the operating life of the transformer is mostly designed to be more than 20 years [14], methanol and ethanol dissolved in oil during operation require to be monitored. The traditional method is to measure the concentration by sampling the transformer oil. With the rise in artificial intelligence prediction methods, it is possible to predict the alcohol concentration in transformer oil. The prediction of alcohol concentration based on intelligent algorithms can track changes in the concentration of alcohols in oil as well as help discover potential risks of transformers in time. Hence, the utilization of intelligent algorithms to predict the concentration of the alcohols is meaningful for the state evaluation and fault prediction of the transformer.

Reviewing existing research, most of the studies focused on the concentration prediction of dissolved gases in oil [15]. Lu et al. proposed the calculation of the gray correlation coefficient of gas feature selection based on gray correlation analysis (GRA) and then used Gaussian process regression (GPR) to predict the dissolved gas value. The results of artificial neural network (ANN), support vector machine (SVM), least squares support vector machine (LS-SVM), and GPR were compared, and it was concluded that the GRA method is more accurate [16]. Zheng et al. reported a dissolved gas prediction method for oil-immersed power transformers that combines wavelet technology and least squares support vector machine (W-LSSVM). The prediction results show that W-LSSVM has a good learning ability for actual limited samples based on the mutation particle swarm algorithm, and the prediction ability is more stable [17]. Then, a combined prediction model based on root function neural network, backpropagation neural network, two different kernel functions of least square support vector machine, and gray model was studied in the literature [18], which can accurately predict dissolved gas in oil. Moreover, Ghunem et al. [19] proposed a prediction model based on transformer oil parameters (breakdown voltage, moisture content, and acidity) and dissolved gas in oil as an input to predict furan content in transformer oil. The prediction model can be adjusted by selecting the most significant predictor and using stepwise regression; then, the verification results showed that the accuracy of the prediction model for furan content in oil can reach 90%. Afterward, Shaban et al. used k-nearest neighbors (k-NNs) as a classification model to classify the furfural data of 731 field transformers and utilized the packaging method as a feature selection method, and a recognition rate of 90% was achieved [20]. In addition, machine learning is also rapidly emerging in polymer science and technology, providing important support for the design [21,22], thermal stability [23], surface area, and crystallinity [24] of polymer materials.

The above studies provide valuable references for predicting the concentration of aging products of transformer insulation. However, methanol and ethanol are indicators of transformer paper insulation aging, and the current research on predicting the concentration of methanol and ethanol in oil based on intelligent algorithms is still lacking. Moreover, the sampling time of the field transformer is irregular, the sample size data are small, and the error of the traditional nonlinear fitting is relatively high. In this situation, this study reports a method for predicting the concentration of methanol and ethanol dissolved in oil. First, the concentrations of methanol and ethanol in the oil-paper insulation at different aging stages are measured. Then, the GA algorithm is utilized to obtain the optimal kernel parameters and penalty factors of the prediction model. Furthermore, SVM is used to perform regression prediction on the test set samples. Finally, the prediction results of several algorithms are compared, which proves the feasibility of GA-SVM for predicting the concentration of methanol and ethanol. This study provides a novel idea for the concentration prediction of methanol and ethanol, which in turn serves to characterize the aging condition of paper insulation.

## 2. Alcohols Generation and Experimental Setup

### 2.1. Generation of Methanol and Ethanol

Generally, the composition of Kraft paper includes 90% cellulose, 6–7% hemicellulose, and 3–4% lignin [25,26]. The molecular structure of cellulose chains is shown in Figure 1.

Jalbert et al. studied the degradation kinetic models of standard wood sulfate and thermally upgraded insulating paper, which further confirmed that methanol is derived from the cut-end chain of cellulose [27]. Generally, during the aging of paper under laboratory conditions of less than 210 °C, the concentration of methanol in transformer oil is always higher than that of ethanol, and the generation rate of ethanol is lower than that of methanol, but it is more stable at the same aging time. It is reported that the main way of methanol production is not pyrolysis. Under acidic hydrolysis conditions, the production of methanol will increase. Ethanol is a molecule found together with methanol in transformer samples. It is produced during the aging process of levoglucosan in oil, its quantity is higher than that of methanol, and it only appears at higher pyrolysis temperatures.

Zhang et al. [28] utilized the molecular dynamics simulation method combining ReaxFF and Monte Carlo to study the formation mechanism of methanol during the degradation of cellulose insulating materials at the atomic or molecular level. The results showed that there are three main ways for the formation of methanol during the degradation of cellulose insulating materials. Liu et al. [29] studied the generation path of ethanol at the atomic level through a series of ReaxFF-molecular dynamics (MD) simulations. The results show that (1) through molecular trajectory analysis, ethanol is mainly derived from vinyl alcohol, an intermediate product of pyrolysis of cellobiose. Then, vinyl alcohol reacts with other groups to produce ethanol. Hence, the production of ethanol requires a secondary reaction of intermediate products. (2) In the early stage of pyrolysis of cellobiose, stable ethanol is not produced. The generation of ethanol is stable and exists throughout the middle and late stages.

### 2.2. Sample Preparation and Parameters Measurement

The detailed information on oil-paper insulation materials selected in this experiment is shown in Table 1.

The pretreatment procedure of oil-paper insulation was according to our published literature [30]. After vacuum drying and oil immersion, the initial moisture in the oil and paper insulation was controlled at 10 mg/kg and 0.8%, respectively. To obtain samples with a similar degree of aging during the service of the transformer, an accelerated thermal aging experiment was conducted on oil-paper insulation at 140 °C for 12 days. It should be noted that the oil/paper mass ratio in this experiment was close to 10:1, and the sampling interval of oil-paper samples was 1 day. In addition, after each oil sample was taken, the corresponding paper sample was also taken out to keep the oil/paper ratio constant.

Afterward, the concentration of methanol and ethanol were measured by gas chromatography–mass spectrometer (GC–MS), and the sampling method was headspace sampling. In this experiment, Shimadzu GC-MS QP2010 was used, and the sample was injected through the DANI automatic headspace sampler (DANI HSS-86.50 PLUS). The test principle of HS–GC–MS is shown in Figure 2. The concentrations of methanol and ethanol in this paper were the average of three measurements.

## 3. Prediction Model of Methanol and Ethanol Concentration Based on GA-SVM

### 3.1. Support Vector Machine Regression Model

For a concentration prediction problem, the historical data are {(*x_i_*, *y_i_*)}, *I* = 1,2,…,*n*, where *n* represents the size of the training sample, *x_i_* represents the characteristics of the sample, and *y* represents the actual concentration of the sample. Hence, the high-dimensional feature space regression equation of the SVM can be expressed as
(1)f(x)=wx+b
where *f*(*x*) is the output, *ω* is the hyperplane normal, and *b* is the offset constant.

According to the judgment standard of generalization ability, in order to make the model better predict the nonlinear data and compare it with the training results, the radial basis kernel function (RBF) is utilized as the kernel function *K*(*x_i_*, *x_j_*) of the SVM model [31], as shown in Equation (2). Here, *g* is the kernel function parameter and satisfies *g* > 0.
(2)$$K(x_{i},x_{j})=exp(-g{\left\|x-g\right\|}^{2})$$

Introducing the concept of statistical learning theory and Vapnik–Chervonenkis (VC) dimension structure risk minimization, the mathematical description of the optimal classification problem can be transformed into the solution of the optimal problem. Therefore, its expression is
(3)$$\left\{ \begin{array}{l} min\varphi (\omega )=\frac{1}{2}{\left\|\omega \right\|}^{2}\\ s.t.\;y_{i}(\omega \cdot x_{i}+b)\geq 1\;\;\;\;\;\;i=1,2,\cdot \cdot \cdot ,n \end{array}\right.$$

After Lagrangian function transformation, and introducing a kernel function to map it to a high-dimensional feature space, the optimal classification surface is determined.

For linear regression problems, the nonlinear regression problem is solved by the same transformation and the introduction of the kernel function to map to the high-dimensional space. Then, the equivalent dual form of the optimization problem is
(4)$$\left\{ \begin{array}{l} \max W(\alpha ,\;\alpha *)=\frac{1}{2}\sum_{i=1}^{n}(\alpha_{i}-\alpha_{i}*)(\alpha_{j}-\alpha_{j}*)K(x_{i}\cdot x_{j})+\\ \;\;\;\;\;\;\;\;\;\;\;\;\;\;\;\;\;\;\;\;\;\;\;\;\;\;\;\;\;\sum_{i=1}^{n}(\alpha_{i}-\alpha_{i}*)y_{i}-\sum_{i=1}^{n}(\alpha_{i}+\alpha_{i}*)\varepsilon \\ s.t.\sum_{i=1}^{n}(\alpha_{i}-\alpha_{i}*)=00\leq \alpha_{i},\alpha_{i}*\leq Ci=1,2,\cdot \cdot \cdot ,n \end{array}\right.$$

In the optimal solution of Equation (4), most of *α_i_* and *α_i_** are zero, and the corresponding sample when *α_i_* or *α_i_** is not zero is the support vector. *α*_1_ and *α_i_** are the introduction of non-negative Lagrange multipliers, and *C* represents the penalty factor. *K*(*x_i_*, *x_j_*) is a kernel function that satisfies Mercer’s condition, which is defined as *K*(*x_i_*, *x_j_*) = *φ*(*x_i_*)·*φ*(*x_j_*). Thus, the expression of the support vector machine regression function can be obtained as follows:(5)$$f(x)=\sum_{i=1}^{n}\left(\alpha_{i}-\alpha_{i}*)K(x_{i},x_{j}\right)+b$$

Moreover, the schematic diagram of the regressive SVM structure is shown in Figure 3. The output in the graph (concentration of methanol and ethanol *C_p_*) is a linear combination of intermediate nodes, and each intermediate node corresponds to a support vector. *x*_1_, *x*_2_, …, *x**_n_* are input variables, and *α_i_* − *α_i_* * are network weights.

It is worth emphasizing that, in this study, four historical data were used as input to predict the next data, so as to ensure the accuracy of the prediction results. The detailed process is shown in Figure 4. In Figure 4, T*_i_*(*i* = 1,2,…, *n*) is the input feature value, ${\hat{T}}_{i}$(*I* = 5–8) is the predicted value of concentration.

Furthermore, in order to eliminate the error caused by large data variation, the sample data are normalized by Equation (6), where *x_min_* and *x_max_* represent the minimum and maximum values of the sample features, respectively.
(6)$${\hat{x}}_{i}=\frac{x_{i}-x_{\min}}{x_{\max}-x_{\min}}$$

The assumptions included in this paper deserve to be noted and can be summarized as follows:
(a)It was assumed that the methanol and ethanol concentrations in the oil are not affected by measurement errors;(b)It was assumed that the methanol and ethanol concentrations in the oil are not affected by external environmental factors;(c)It was assumed that the methanol and ethanol concentrations at different aging stages correspond to the concentrations during the transformer operation for 0–35 years.


### 3.2. Optimization of Hyperparameters of SVM Based on GA Algorithm

Parameters *C* and *g* directly affect the predictive ability and algorithm efficiency of the regression prediction model and affect the robustness of the regression prediction model. When the RBF kernel function is used, the influence of the kernel function *g* is derived from the radial basis function neural network. The larger the *g*, the stronger the influence between the support vectors, which is likely to cause under-learning. Conversely, the smaller the *g* value is, the easier it is to overlearn, and the generalization ability becomes worse. Given this, a genetic algorithm has the characteristics of fast and global optimization. Thus, the genetic algorithm was used in this paper to optimize the penalty factor *C* and the kernel parameter *g* of the SVM model. The fitness function in this study needs to use the idea of cross-validation (CV), which divides the original sample into the training set and validation set. Thus, the validity of the model can be verified through the verification set.

In this paper, parameters *C* and *g* use binary coding, meaning that each chromosome is composed of two genes (*C* and *g*). Moreover, the selection operator uses random traversal sampling, and the crossover operator uses a single point crossover operator.

To evaluate the model more effectively, in this paper, fivefold cross-validation was used to obtain the average mean square error (*MSE*) of the model established under different parameter selections, which was used as the evaluation standard for the quality of the model established under the parameter selection. By finding the smallest average *MSE* under cross-validation, the optimal parameters were found. The process is looped *k* times, and the average of the *MSE* of these *k* times is taken as the fitness function of the optimization process, and the expression is
(7)$$\mathrm{MSE}=\frac{1}{n}\sum_{i=1}^{n}{\left[f(x_{i})-y_{i}\right]}^{2}$$

Among them, *n* is the number of samples in the validation set, *f*(*x_i_*) and *y_i_* are the predicted concentration and actual concentration of the *i*th test sample, respectively.

### 3.3. The Prediction Steps of the GA-SVM Model

The parameter setting information of the GA algorithm is shown in Table 2. Additionally, the concentration prediction process of the GA-SVM is described in Figure 5. Furthermore, the GA optimization for the penalty factor *C* and the kernel parameter *g* of SVM mainly includes the following steps:
(1)The penalty factor *C* and kernel function parameter *g* were initialized, and binary coding was utilized to encode *C* and *g*;(2)Various parameters of the GA algorithm were set according to Table 2;(3)SVM training was performed on the initial population, and the fitness of the individual was calculated according to the recognition rate of the training samples;(4)The selection, crossover, and mutation operations on parameters were performed according to individual fitness to obtain a new generation of populations. Then, SVM training was performed on the new population to calculate individual fitness;(5)Fitness was assessed by checking whether the population satisfied the termination condition. If the termination condition was met, the individual with the greatest fitness was output as the optimal parameter, and the optimal parameter was used for prediction. Otherwise, the evolutionary algebra was increased, and the process was repeated from step 4 to continue running the program;(6)The measured and predicted values of methanol and ethanol concentrations were compared, and the *MSE* of the corresponding prediction models were obtained.


## 4. Prediction Process of the Concentration of Methanol and Ethanol

### 4.1. The Prediction Results of GA-SVM

Based on the GA-SVM algorithm, through fivefold cross-validation, the concentration of methanol and ethanol were optimized. Additionally, the optimal values for penalty parameter *C* and kernel function *g* were obtained. The maximum number of selected populations was 200, and the maximum number of evolutionary generations was 200. The fitness curve of the GA algorithm is shown in Figure 6.

As shown in Figure 6, in the initial stage of genetic evolution, the fitness level drops sharply. Obviously, as the evolutionary algebra increases, the fitness eventually remains stable. Furthermore, the relationship between the penalty factor *C*, the kernel function parameters *g*, and the corresponding *MSE* is depicted in Figure 7. It can be seen that when the values of *C* and *g* are too large or too small, the model will be underfitting, whereas when the values of *C* and *g* are too large, the model will be overfitting.

The methanol and ethanol concentration predictions in this study were based on a Core i9 octa-core processor i9-9900K. The basic frequency of the processor is 3.6 GHz, and the memory is 32 GB. In addition, the optimal parameters, mutation probability, and *MSE* results selected by the GA-SVM prediction model are listed in Table 3.

### 4.2. Comparison of Prediction Results of Different Intelligent Algorithms

In this section, the accuracy and usability of GA-SVM for predicting the concentration of methanol and ethanol are discussed. The prediction results based on GA-SVM, backpropagation neural networks (BPNN), decision tree, random forest, Bayesian-SVM, Adaboost, and linear regression were compared. The detailed prediction information is listed in Table 4. In addition, the comparison of the prediction results of different algorithms is shown in Figure 8. It is worth noting that the prediction results and errors of methanol and ethanol concentrations in this paper resulted from the average value of the algorithm running 30 times. Moreover, the learning rate of BPNN was 0.01, the activation function was ReLU, the epoch was 100, and the dropout was 0.8.

As illustrated in Figure 8a, the accuracy of the methanol concentration prediction results based on GA-SVM is higher than other algorithms, which is closer to the measured result. The prediction result of the backpropagation neural network (BPNN) algorithm is the worst, which may be related to its insufficient network structure design, learning algorithm, and convergence effect. The prediction result of the random forest algorithm is much higher than that of GA-SVM. Random forest is an ensemble algorithm of regressor based on decision trees, which can avoid overfitting problems in decision trees. As evident from Figure 9b, the ethanol concentration prediction results based on GA-SVM are also the most accurate. The prediction results based on decision trees and random forests are very different from those of GA-SVM. The accuracy of Bayesian-SVM is relatively good, which shows that the SVM model is better than other predictive models. Nevertheless, in terms of parameter optimization, GA can find the optimal parameters better than the Bayesian algorithm; thus, GA-SVM has the best prediction performance. Moreover, the methanol prediction result of the Adaboost algorithm is better than that of random forest, but due to its dependence on weak learners and small samples, the accuracy is not as high as that of SVM. In addition, the results of linear regression prediction show that several prediction points are close to the actual measured value, but due to the unstable prediction, the accuracy of several prediction points is poor.

Meanwhile, the relative errors between the predicted results and measured results of methanol and ethanol based on different algorithms were calculated. The results are shown in Figure 9.

Noticeably, from Figure 9, with the exception of the decision tree and random forest algorithm, the error of the methanol concentration prediction results of other algorithms increases with the increase in aging time. The error of GA-SVM is stable within 10%, and the prediction error of other algorithms is much greater than that of GA-SVM. The error of the GA-SVM is relatively close to the Bayesian-SVM and linear regression algorithm. The errors of the two algorithms are relatively close, but the prediction error fluctuates within a certain range. The prediction error of ethanol concentration further shows that GA-SVM is the most reliable compared with other algorithms. Although the prediction results of Bayesian-SVM in the early stage and GA-SVM are similar, the performance in the mid-stage is not good. Thus, the above results prove that the GA-SVM model for predicting methanol and ethanol concentrations proposed in this research is acceptable.

Additionally, to compare the prediction effects of different algorithms more intuitively, the *MSE* of methanol and ethanol concentration prediction is shown in Figure 10. It can be clearly seen that, compared with other algorithms, GA-SVM has the smallest error in predicting alcohol concentration. In other words, GA-SVM has high accuracy in predicting methanol and ethanol concentration. In order to better demonstrate the study process in this paper, the framework flowchart is shown in Figure 11.

## 5. Software Construction of Prediction Model

The above-mentioned concentration prediction model can provide a basic tool to assess the operating state of the transformer. Moreover, an industrial software program for concentration prediction of methanol and ethanol was developed. The designed software interfaces are shown in Figure 12.

As shown in Figure 12, first, methanol or ethanol data were imported, and the software read the data; then, the parameters of the prediction model were set. For ease of operation, a module was specially set up. This module sets the quantity of data in the previous aging stages and then predicts the next data. The parameter settings include the number of population iterations, the number of populations, the crossover probability et al. The above parameter values can be adjusted according to each forecast demand. Furthermore, after clicking the “Run” button, the corresponding *MSE* of different penalty factors and kernel parameters, and the comparison image of the predicted value and the true value were output. Finally, the error between the predicted value and the true value was also output. The user can also click the “Data Storage” button to export the forecast data for the next review. It should be noted that when new experimental data were obtained in the later stage, the relevant parameters were directly adjusted through the software to perform the prediction process.

## 6. Conclusions

In this paper, a genetic-algorithm-optimized support vector machine was applied to the prediction of the concentration of methanol and ethanol dissolved in oil for the first time. The main conclusions obtained are as follows:
(1)The average *MSE* of methanol concentration prediction based on GA-SVM reached 0.008, and the average *MSE* of ethanol concentration prediction reached 0.003. The undeniable limitation is that a large volume of experimental data is difficult to obtain, but the proposed method in this research has a higher prediction accuracy and can be applied to the concentration prediction of methanol and ethanol;(2)The predicted concentrations of methanol and ethanol in the prediction model were obtained based on the data of the first four samples, which improved the accuracy of the prediction. The optimization of parameters had a significant impact on the prediction results of SVM. The GA-SVM prediction model introduced in this paper can overcome the blindness of the parameter selection of the traditional model;(3)The prediction accuracy of the oil alcohol prediction model studied in this paper was acceptable. In any case, the experimental sample data were small, and some prediction errors were inevitable, but they can provide points of reference for the prediction of alcohol concentration. In further research, long-term aging experiments will be conducted, and more data will be obtained for concentration prediction;(4)Additionally, according to the prediction method of alcohol concentration based on GA-SVM, an industrial software program was developed to facilitate the later methanol and ethanol concentration prediction and parameter adjustment.


Currently, the proposed model in this paper was obtained based on the data under laboratory conditions. In order to obtain a prediction model with high generality and accuracy, in future research, we will track the concentration of alcohols in a certain number of transformers under field conditions.

## Figures and Tables

**Figure 1 polymers-14-01449-f001:**
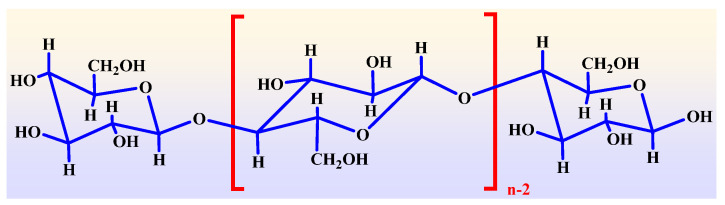
Molecular structure of cellulose in insulating papers.

**Figure 2 polymers-14-01449-f002:**
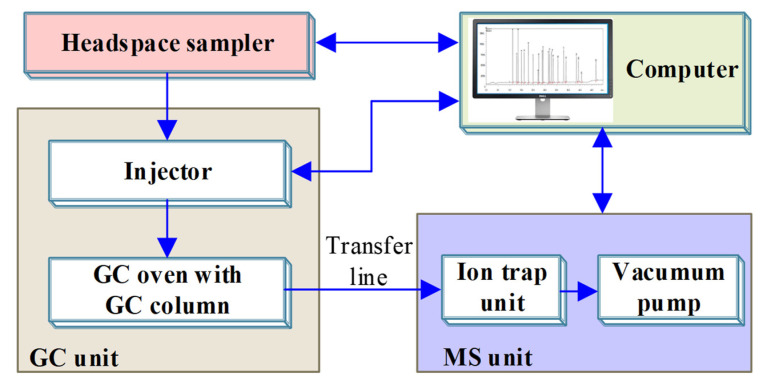
The schematic diagram of HS–GC–MS.

**Figure 3 polymers-14-01449-f003:**
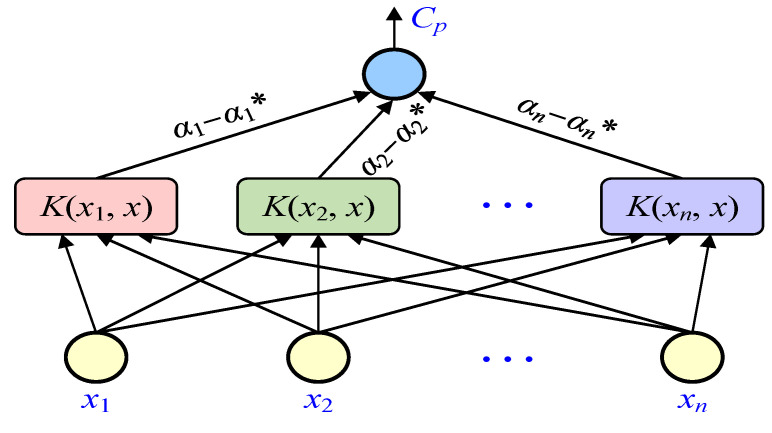
Structure diagram of SVM regression.

**Figure 4 polymers-14-01449-f004:**
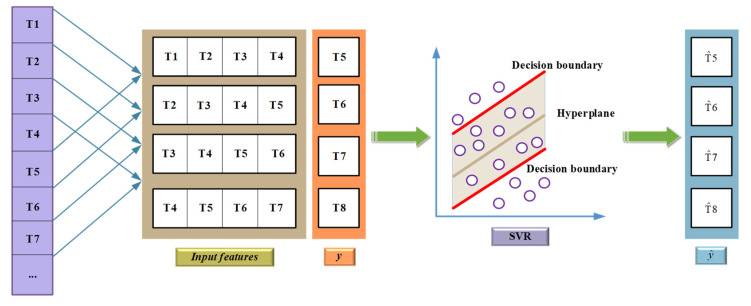
Prediction process of SVM regression.

**Figure 5 polymers-14-01449-f005:**
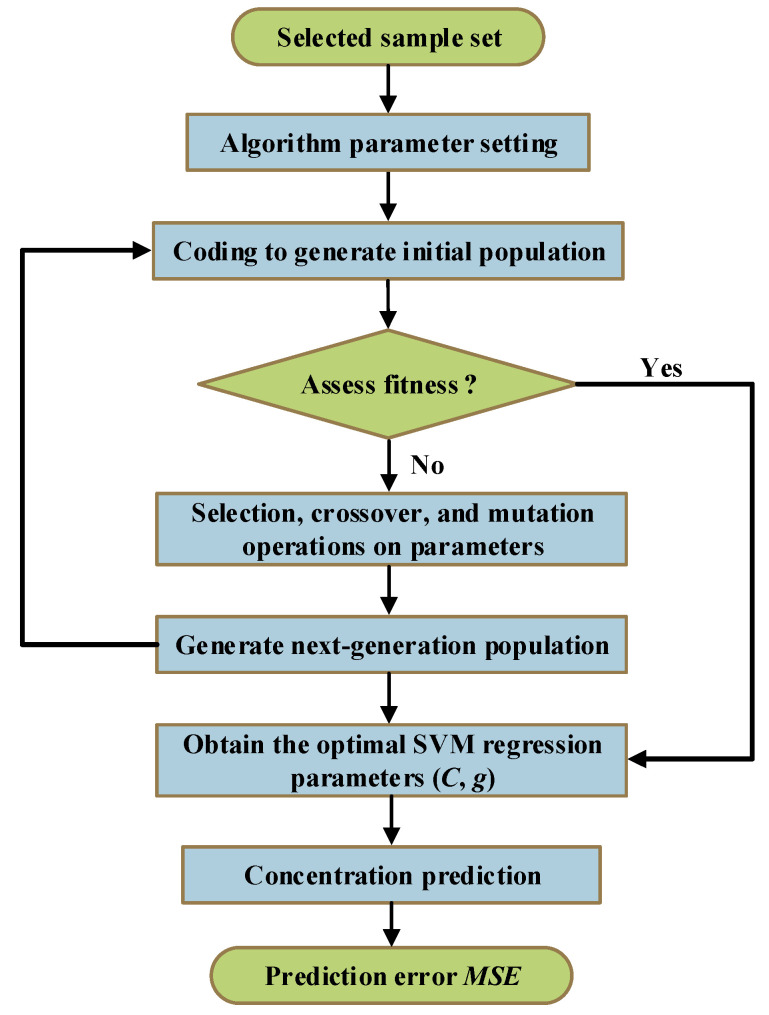
Modeling flowchart of GA-SVM.

**Figure 6 polymers-14-01449-f006:**
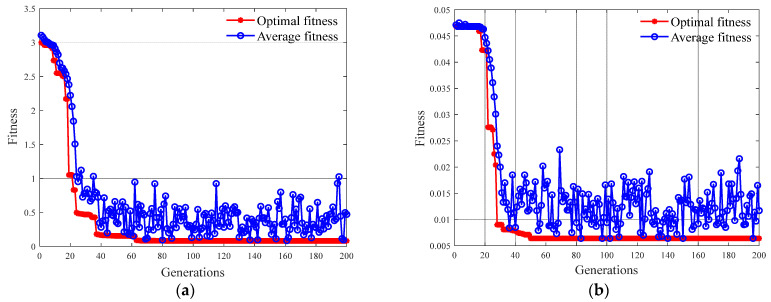
Fitness curve optimized by GA: (**a**) methanol; (**b**) ethanol.

**Figure 7 polymers-14-01449-f007:**
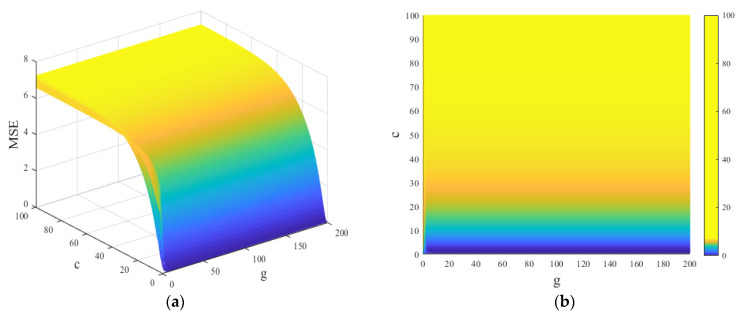
Accuracy for sample points and the optimal point that GA found: (**a**) 3D; (**b**) 2D.

**Figure 8 polymers-14-01449-f008:**
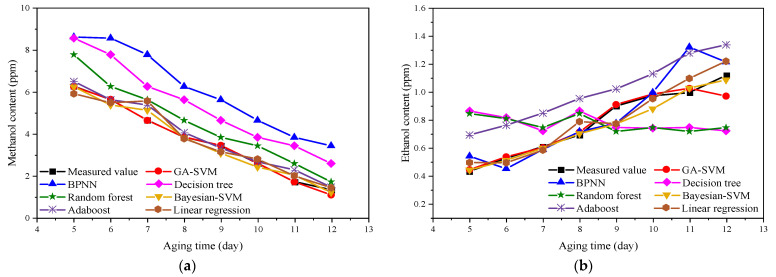
Comparison of prediction results of different algorithms: (**a**) methanol; (**b**) ethanol.

**Figure 9 polymers-14-01449-f009:**
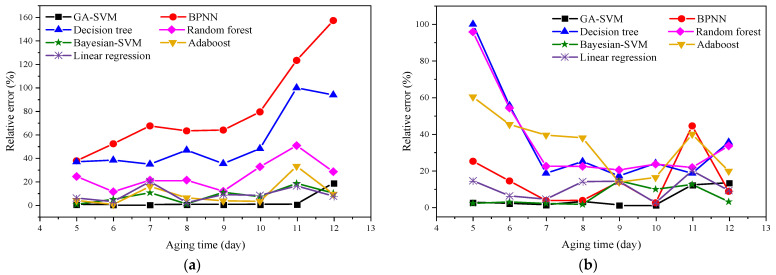
Relative error results of different algorithms: (**a**) methanol; (**b**) ethanol.

**Figure 10 polymers-14-01449-f010:**
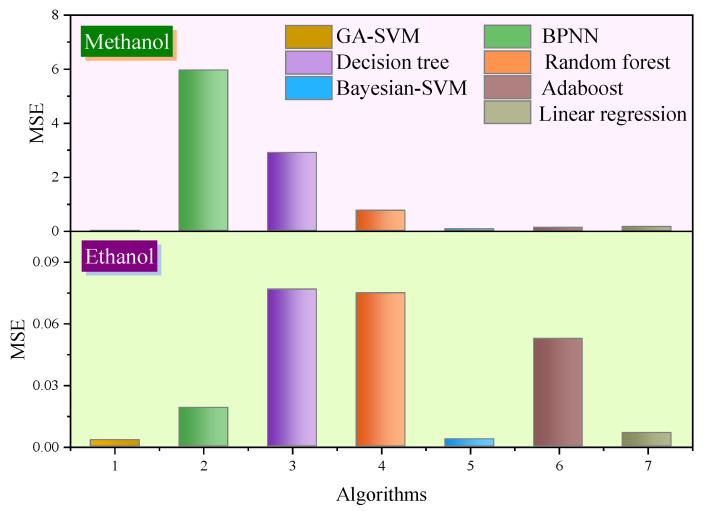
*MSE* for concentration prediction of methanol and ethanol.

**Figure 11 polymers-14-01449-f011:**
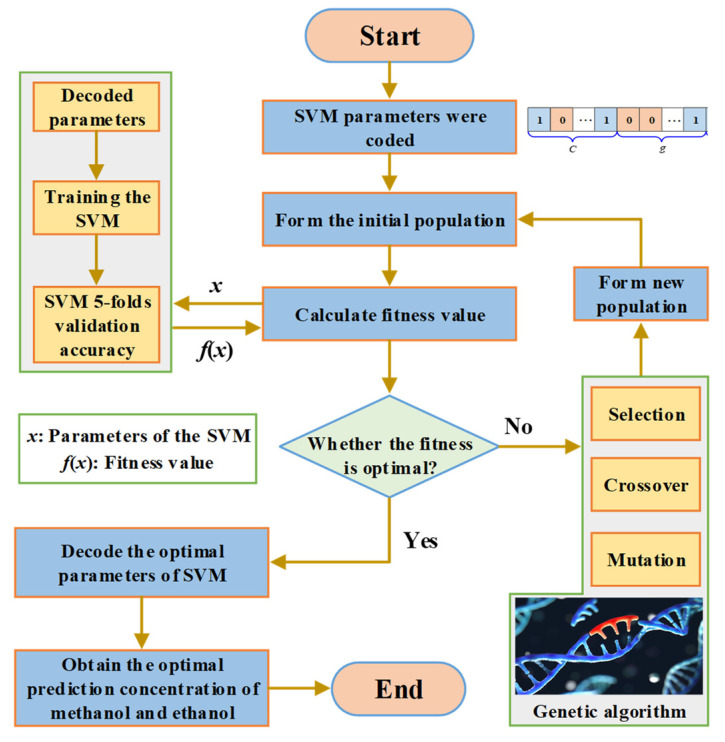
The algorithm framework diagram.

**Figure 12 polymers-14-01449-f012:**
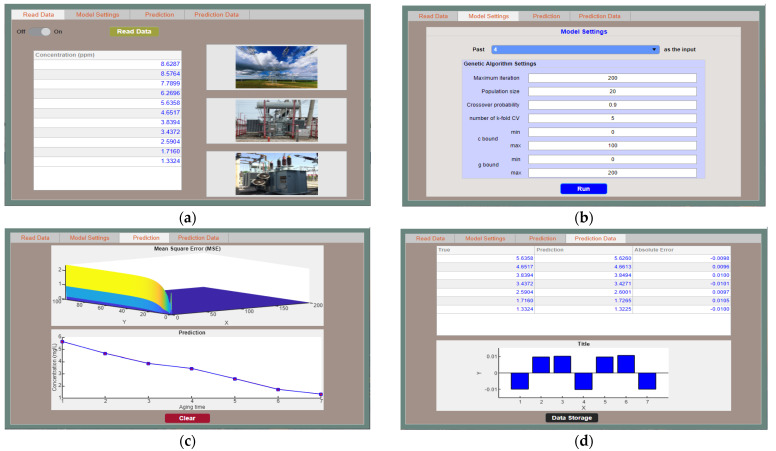
The interface of concentration prediction software: (**a**) read data; (**b**) parameters setting; (**c**) prediction plot; (**d**) prediction data.

**Table 1 polymers-14-01449-t001:** The specific parameters of insulation materials.

Materials Type	Parameters	Value
Insulating oil	tanδ	4 × 10^−4^
	Pour point	≤−45 °C
	Flash point	135 °C
	Manufacturer	Sinopec Group Co. Ltd., Chongqing, China
Insulating paper	Thickness	1 mm
	TS	MD: 138.79 MPa, CMD: 97.14 MPa
	Density	1.09 g/cm^3^
	Manufacturer	Weidmann high voltage insulation Co. Ltd., Taizhou, China

**Table 2 polymers-14-01449-t002:** The detailed parameters settings of GA.

Parameters	Settings
Maximum iteration	200
Population size	20
Mutation probability	0.03
Crossover probability	0.9
Range of penalty factor *C*	(0, 100)
Range of kernel function parameter *g*	(0, 1000)

**Table 3 polymers-14-01449-t003:** The output parameters of GA-SVM.

Parameter	Best *C*	Best *g*	Mutation Probability	*MSE*
Methanol	50.7794	0.5016	0.0832	0.0780
Ethanol	68.1133	0.7896	0.0212	0.0034

**Table 4 polymers-14-01449-t004:** The prediction results of alcohols concentration based on different algorithms.

Indicator	Aging Time (day)	Measured Value	GA-SVM	BPNN	Decision Tree	Random Forest	Bayesian-SVM	Adaboost	Linear Regression
Methanol	5	6.2696	6.2506	8.6207	8.5684	7.7820	6.2511	6.5027	5.9141
	6	5.6358	5.6466	8.5684	7.7820	6.2617	5.3741	5.6257	5.4871
	7	4.6517	4.6425	7.7820	6.2617	5.6278	5.1281	5.3797	5.5711
	8	3.8394	3.8507	6.2617	5.6278	4.6438	3.8121	4.0637	3.7701
	9	3.4372	3.4481	5.6278	4.6438	3.8314	3.0731	3.3247	3.1411
	10	2.5904	2.6009	4.6438	3.8314	3.4292	2.4131	2.6647	2.7961
	11	1.7160	1.7073	3.8314	3.4292	2.5825	2.0291	2.2807	1.9921
	12	1.3324	1.0849	3.4292	2.5825	1.7081	1.2011	1.4527	1.4301
Ethanol	5	0.4317	0.4429	0.5398	0.8638	0.8455	0.4403	0.6920	0.4939
	6	0.5247	0.5356	0.4499	0.8164	0.8091	0.5102	0.7618	0.4928
	7	0.6089	0.6014	0.5876	0.7215	0.7447	0.5972	0.8488	0.5830
	8	0.6915	0.7125	0.7165	0.8638	0.8455	0.7016	0.9532	0.7876
	9	0.8995	0.9099	0.7749	0.7461	0.7170	0.7705	1.0221	0.7727
	10	0.9736	0.9848	0.9961	0.7407	0.7450	0.8792	1.1308	0.9518
	11	0.9950	1.0261	1.3216	0.7461	0.7170	1.0281	1.2797	1.0971
	12	1.1191	0.9704	1.2150	0.7215	0.7447	1.0871	1.3387	1.2201

## Data Availability

The data presented in this study are available on request from the corresponding author.

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
