# Peer review of "Concentration Prediction of Polymer Insulation Aging Indicator-Alcohols in Oil Based on Genetic Algorithm-Optimized Support Vector Machines"

_polymers, 2022, doi:10.3390/polym14071449_

Round 1

Reviewer 1 Report

Authors have presented their work properly. They have given comparison of GA-SVM with other algorithms and their conclusions are supported by presented results.

Author Response

Point 1:

Authors have presented their work properly. They have given comparison of GA-SVM with other algorithms and their conclusions are supported by presented results.

Response 1:

Thank you for your valuable comments and recognition of our work. Our work aims to predict the concentration of methanol and ethanol dissolved in transformer oil using GA-SVM algorithm. Finally, the prediction results are compared with other algorithms. The results show that the prediction model used in this paper is more accurate, and can well track the changing trend of methanol and ethanol concentration in transformer oil. In order to obtain a prediction model with high generality and accuracy, in future research, we will track the concentration of methanol and ethanol in a certain number of the transformer under field conditions.

We would like to express our gratitude once again!

Reviewer 2 Report

Dear Authors,

The ideas in the manuscript are interesting. Using software to predict properties is certainly very important. However, some minor changes are needed to make the manuscript easier to read.

The overall English in the manuscript needs some improvement. There are many grammatical errors and usage of wrong tense through the manuscript. References are not consistently cited throughout this manuscript

I have pointed out a few grammatical errors for your reference.

  • Introduction- Please rephrase and correct all grammatical errors to make the flow better.
  • In the introduction, there are several instances where the tense is not consistent, please make it consistent.
  • Line 61 and 67-Please quote author name instead of literature 16 and 17
  • Line 75, please include the reference for the paper you are referencing here.
  • Line 102-106, can you please clarify? Looks like these lines don’t belong here
  • Line 120, please cite the reference properly

Reviewer 3 Report

The manuscript is of good quality, it is relatively well-written, although the background could be more general to place the work in better context. The results are new and of interest to various researchers as the topic is highly interdisciplinary. Some details need to be clarified and improved before further consideration. Major revision is needed before any further consideration.

  1. The abstract should mention the potential impact of the research work presented in the manuscript. What is the novelty and how the results will influence the subject field?
  2. Justification for the selected ML algorithm should be provided. What are the other algorithms that could have been employed?
  3. The used dataset and model are not provided in a usable format. Without an available dataset, the external validation of the reported models cannot be performed. Provide the used dataset as an external file (.json etc) as well.
  4. Provide a downloadable format of the models that can be directly used by the community in the future, which is one of the main goals of the work.
  5. For small datasets (such as this work), simple machine learning algorithms (besides SVM: ANN, boosting algorithm, etc.) can perform extremely well. Comparing the obtained results could be an additional layer of validation. Include the mentioned machine learning algorithms, optimize their performance and compare the results.
  6. All the assumptions should be systematically listed in the manuscript.
  7. Validation, overfitting and the effect of sample numbers should be investigated in depth.
  8. A more thorough cross-validation would be appreciated and the model is lacking some repeatability measures such as y-scrambling. Without this, the model is subjected to random fluctuation, which occurs later. Also, the model should run at least 30 times (using different random seeding each time) at each hyperparameter state and report the average results and error.
  9. Machine learning in polymer science and technology is emerging and this should be mentioned briefly with broad examples listed (10.1039/D1GC02796D; 10.1021/acsapm.1c00486; 10.1039/D0GC02956D; 10.1021/acs.chemmater.1c02061).
  10. The hyperparameter tuning is not clearly described in the model building. Only a subset of parameters are varied? Include a proper hyperparameter tuning: including the optimization of the learning rate, activation function (sigmoid, ReLU, tanh, etc.), cross-validation (see comment above), split size, optimization algorithm (adam, gradient descent, etc.), cost, and loss function, number of activation units, drop-out, number of epoch (or any other hyperparameter, that must be important).
  11. A more detailed mathematical framework should be added to the manuscript. This is necessary to ensure that the general readership of ‘polymers’ will understand the work. The authors write to a non-specialized audience and therefore a more thorough background should be provided.
  12. The transferability of the results should be discussed. The presented results are specific to the studied system, i.e. case study, or can be applied to other systems?

Round 2

Reviewer 3 Report

The authors have done a very thorough revision.

This manuscript is a resubmission of an earlier submission. The following is a list of the peer review reports and author responses from that submission.